# Exploring noise annoyance and sound quality for airborne wind energy systems: Insights from a listening experiment

Helena Schmidt[1], Renatto M. Yupa-Villanueva[2], Daniele Ragni[1], Roberto Merino-Martínez[2], Piet van Gool[3], Roland Schmehl[1]

[1]Department of Flow Physics and Technology, Delft University of Technology, Delft, 2629 HS, The Netherlands

[2]Department of Aircraft Noise and Climate Effects, Delft University of Technology, Delft, 2629 HS, The Netherlands

[3]Department of Industrial Engineering and Innovation Sciences, Eindhoven University of Technology, Eindhoven, 5612 AE, The Netherlands

*Correspondence to*: Helena Schmidt (h.s.schmidt@tudelft.nl)

**Abstract.** This study investigates the relationship between sound quality metrics (SQMs) and noise annoyance caused by airborne wind energy systems (AWESs). In a controlled listening experiment,75 participants rated their annoyance on the International Commission on Biological Effects of Noise (ICBEN) scale in response to recordings from in-field measurements of two fixed-wing and one soft-wing ground-generation AWES. All recordings were normalized to an equivalent A-weighted sound pressure level of 45 dBA. The results revealed that sharpness was the only SQM predicting participants' annoyance. Fixed-wing kites, characterized by sharper and more tonal and narrowband sound profiles, were rated as more annoying than the soft-wing kite, characterized by higher loudness values. In addition, the effect of some SQMs on annoyance depended on participant characteristics, with loudness having a weaker impact on annoyance for participants familiar with AWESs and tonality having a weaker effect on annoyance for older participants. These findings emphasize the importance of considering psychoacoustic factors in the design and operation of AWESs to reduce noise annoyance.

## 1 Introduction

Wind energy is one of the most widely available renewable energy sources, and its capacity must increase by 320 GW by 2030 to meet the climate goals of the Paris Agreement (IEA, 2023; UNFCCC, 2016). A promising yet unexploited novel renewable energy technology is airborne wind energy (AWE) (BVG Associates, 2022; Vos et al., 2024). AWE uses tethered flying devices, called kites, to harness higher-altitude winds. AWE can complement conventional wind energy by accessing stronger, more consistent wind resources above 200 meters and providing power in remote or temporarily used locations, such as in the aftermath of natural disasters. With its substantially lower mass compared to conventional wind turbines, AWE also has a smaller environmental footprint (Hagen et al., 2023). While AWE is regarded as a potential game-changer for the energy

transition because it can harness higher-altitude winds, requires fewer materials compared to wind turbines, and can be deployed in remote or distant offshore locations (IRENA, 2021), the technology has not yet converged towards a single
standard configuration.

The existing prototypes can be divided into two main configurations: ground-generation and fly-generation concepts, as shown in Fig. 1 (Cherubini et al., 2015). The former concept alternates between energy-generating reel-out and energy-consuming reel-in phases. During the reel-out phases, the kite is flown in a loop or figure of eight maneuvers, generating more energy than is used during the reel-in phases, resulting in a positive net power output (Vermillion et al., 2021). Ground-
generation AWE systems (AWESs) commonly use soft-wing kites based on flexible membrane wings or fixed-wing kites typically made from carbon fiber-reinforced polymers. The latter concept employs small wind turbines onboard a fixed-wing kite to generate electricity directly while airborne.

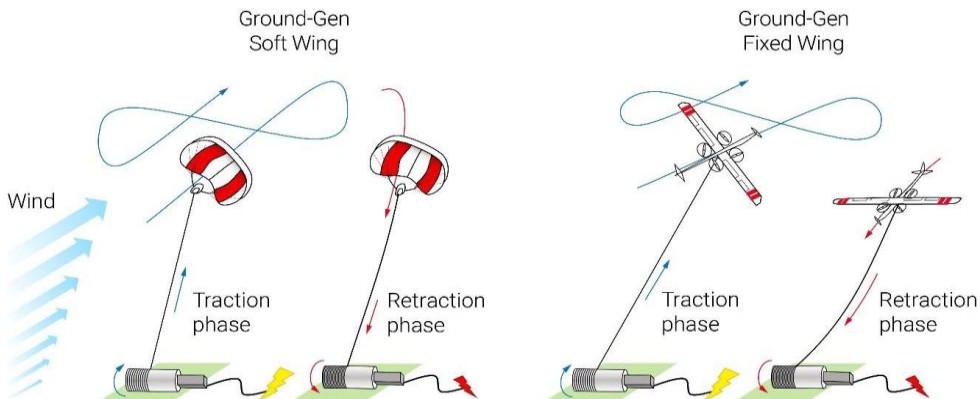

**Figure 1: Schematic representation of ground-generation airborne wind energy systems employing a soft-wing and a fixed-wing kite, respectively** (based on Fagiano et al., 2022)**.**

AWESs, like all wind energy technologies, must comply with environmental regulations on sound emissions to limit the impact on surrounding residents (van Kamp and van den Berg, 2021). Noise is a primary source of public opposition to
wind turbines and a central aspect of debates on their social acceptance (Bednarek-Szczepańska, 2023; Kirkegaard et al., 2024; Taylor and Klenk, 2019). While the health effects of noise remain contentious, even within the scientific community (Kirkegaard et al., 2024; Taylor and Klenk, 2019), substantial evidence indicates that individuals living near wind farms may frequently report noise annoyance, often accompanied by complaints such as sleep disturbances, psychological distress, and general functional impairments (Bakker et al., 2012; Godono et al., 2023; Haac et al., 2019; Hübner et al., 2019; Ki et al.,
2022; Michaud et al., 2016a; Müller et al., 2023; Pawlaczyk-Łuszczyńska et al., 2014; Pedersen and Persson Waye, 2004, 2007; Pohl et al., 2018; Radun et al., 2019; Turunen et al., 2021). Noise annoyance is typically defined as a negative evaluation of wind turbine sound emissions (Pohl et al., 2018) and is influenced by factors such as expected health impacts, perceived fairness in the planning process, individual sensitivity to noise, and the visual and landscape impact of the turbines (Haac et

al., 2019; Hübner et al., 2019; Michaud et al., 2016b; Müller et al., 2023; Pawlaczyk-Łuszczyńska et al., 2018; Schäffer et al.,

2019; Tonin et al., 2016).

Although AWESs are often assumed to be quieter due to their higher operational altitudes (for a review, see Schmidt et al., 2022), this assumption disregards several factors that influence noise perception. These factors include individual dispositions (e.g., noise sensitivity, especially to low-frequency sounds) (Haac et al., 2019; Michaud et al., 2016b; Pedersen et al., 2010; Pedersen and Persson Waye, 2007; Schutte et al., 2007), perceptions (e.g., the aesthetics of the technology or fairness

of the planning process) (Haac et al., 2019; Hübner et al., 2019; Pedersen and Larsman, 2008), and attitudes towards wind energy projects or the technology itself (Hoen et al., 2019; Hübner et al., 2019; Ki et al., 2022; Pawlaczyk-Łuszczyńska et al., 2018; Pedersen and Persson Waye, 2007; Schäffer et al., 2019). Technological aspects also play a role, including tethers, onboard rotating components, and the relatively high speeds at which kites operate, which enhance tonal components and modulation of the sound emitted (Hansen et al., 2021; Lee et al., 2011; Schäffer et al., 2018; Torija et al., 2019; Yokoyama

and Tachibana, 2016; Yonemura et al., 2021).

Although research on AWES sounds is still limited, a preliminary study by Bouman (2023) revealed differences in the noise profiles of fixed-wing and soft-wing kites: The fixed-wing kite had a narrowband spectral distribution of the emitted noise, enhanced by laminar flow regimes on the suction side of the wing with a relatively short chord. The larger soft-wing kite produced a broadband distribution largely determined by turbulent boundary-layer trailing-edge noise. However, how

these noise sources relate to noise annoyance has not been investigated to date. Schmidt et al. (2024) conducted the only field study so far on AWES sound emissions, finding that 35.2% of respondents living on average 2 km from the soft-wing AWES could hear its sounds at home, with 13.1% being annoyed (score of at least 2 on a scale from 0 to 4) and 7.5% highly annoyed (score of at least 3 on the same scale). However, the study did not investigate the relationship between the AWES's sound emissions and the reported annoyance, leaving a critical gap in understanding the impact of AWES noise on communities.

The AWE industry has primarily focused on improving system reliability and scalability, with less emphasis on noise mitigation. However, developers are increasingly recognizing the challenges posed by noise (Junge et al., 2023) and are beginning to develop measurement methods and gather insights to mitigate its effects. Early acoustical research, like the present study, plays a crucial role in identifying factors that contribute to noise annoyance for AWESs. This knowledge can guide the design and implementation of mitigation measures before the technology becomes constrained by fixed design choices.

Existing research on wind turbines has typically relied on conventional sound indicators, such as the equivalent sound pressure level Leq or its A-weighted version $L$p,A,eq (Kephalopoulos et al., 2014; Pieren et al., 2019). The $L$p,A,eq metric adjusts sound measurements to the sensitivity of the human ear, particularly to frequencies between 500 Hz and 6 kHz. However, these metrics do not adequately reflect the sound properties that explain annoyance (Bockstael et al., 2011; Pedersen and Persson Waye, 2004; Persson Waye and Öhrström, 2002; Pieren et al., 2019), such as the tonal and high-frequency content

of turbine noise that has been linked to stronger annoyance (Oliva et al., 2017; Persson Waye and Agge, 2000; Yokoyama and Tachibana, 2016). Similarly, while the effective perceived noise level (EPNL), developed for aircraft noise (U.S. Department of Transportation, 2017), accounts for sound magnitude, spectral content, duration of the sound signal, and tonal components,

it may not be well-suited for evaluating AWES noise (Kryter, 1960). Sound quality metrics (SQMs), such as loudness, tonality, sharpness, roughness, and fluctuation strength, offer an alternative approach by focusing on perceptual aspects of sound (Greco et al., 2023). Research on wind turbines (Merino-Martinez et al., 2019a; Persson Waye and Öhrström, 2002; Pockelé and Merino Martínez, 2024) and aircraft noise (Merino-Martinez et al., 2019b; More, 2010; Pereda Albarrán et al., 2018, 2017; Sahai, 2016; Vieira et al., 2019) has begun to explore how useful SQMs are for understanding annoyance.

The present study aims to investigate how well SQMs predict noise annoyance caused by AWESs. It was not assumed that participants were knowledgeable about specific SQMs, nor were they informed of these metrics during the experiment. Instead, these metrics were objectively derived from acoustic analyses of the recordings. The study also explores Psychoacoustic Annoyance (PA) metrics, which combine multiple SQMs into a single predictor of annoyance, comparing the PA metrics' performance with the conventional metric EPNL. The benefit of PA metrics is that they provide a quick estimate of the noise annoyance perceived for a given sound without measuring respondents' annoyance levels. Additionally, the performance of PA metrics for estimating annoyance levels will be. Using recordings from both soft-wing and fixed-wing kites, this study conducts a controlled listening experiment to assess annoyance ratings for AWES.

Section 2 describes the study design, procedure, and materials. Section 3 presents the results from the acoustical analyses of the sound recordings and the statistical analyses of the reported annoyance. Finally, Section 4 summarizes the key findings and their implications.

## 2 Methodology

In the following, the methodologies employed to record the sound samples and the laboratory listening experiment are explained in detail, including characteristics of the sound samples and participants, annoyance ratings, and laboratory procedures.

### 2.1 Sound recordings

Nine sound recordings from three different AWESs (i.e., three recordings from each prototype) were used for the listening experiment. A total signal length of 25 s per recording was extracted from longer, more complex audio footage that included additional non-relevant preparation phases for the three AWESs. All three AWESs implement ground-based electricity generation (see Section 1). One is a soft-wing kite (AWES A), and the other two are fixed-wing kites (AWESs B and C). Table 1 provides more information about the AWESs and the sound measurement campaigns.

The three recordings for each AWES were chosen to represent typical sound emissions during the reel-out phase of their respective systems. During this phase, the kite operates in crosswind maneuvers at high flight speeds while the reel-out speed is kept relatively low to maximize the energy production period. This operational setup implies that sound emissions from the kite, including contributions from onboard ram-air turbines, wing flutter, and tether vibrations, are the most

significant. In contrast, sound emissions from the ground station (e.g., the generator) are comparatively minor due to the low reeling speed. Due to AWES C being towed by a truck, its recordings exhibited greater variability compared to the more consistent sound profiles of kites A and B.

Given that there are currently no specific sound regulations for AWESs, the sound pressure levels of the recordings were normalized to an equivalent A-weighted sound pressure level value of 45 dBA to align with European regulations for wind turbines, which commonly range between 35 dBA and 55 dBA during the day (Solman and Mattijs, 2021). Normalization refers to adjusting the sound pressure levels of recordings to a common reference value, ensuring comparability. A-weighting is a standard method to adjust sound measurements to reflect the human ear's sensitivity (approximately between 2 and 5 kHz). Additionally, normalizing the sound pressure levels helps to evaluate aspects of sound quality other than loudness (Boucher et al., 2024).

**Table 1** Overview of the investigated airborne wind energy systems (AWESs) and the corresponding sound measurement campaigns.

| | AWES A | AWES B | AWES C |
|---|---|---|---|
| Kite type | Soft-wing | Fixed-wing | Fixed-wing |
| VTOL propellers | None | Present, inactive during the measurements | Present, inactive during the measurements |
| Ram-air turbine | Present, tied down during the measurements to prevent free-spinning | None | Present, active during the measurements |
| Flight pattern | Figure of eight | Circle | Circle |
| Wind speed (m/s) | 5 – 10 | 9 | 8 – 9[a] |
| Max relative flying airspeed (m/s) | 38 | 42 | 43 |
| Max kite altitude during experiment (m) | 253 | 231 | 150 |

| | | | |
|---|---|---|---|
| Distance to microphone (m) | 428 – 620 | 305 – 689 | Approximately 100 – 700 |
| Test location and type | Field; standard flight test | Inoperative airfield; standard flight test | Inoperative airfield; tow test (i.e., the ground station was on the back of a truck driving straight to create an artificial wind field while the kite was flying crosswind loops of about 60 – 70 m diameter) |
| Recording instrumentation | Brüel & Kjær 4189 microphone at 1 m height and 650 m downwind from the winch of the ground station; Brüel & Kjær UA-650 windscreen over the microphone to reduce wind sounds; Brüel & Kjær sound level meter 2250 | Brüel & Kjær 4189 microphone at 1 m height and 679 m downwind from the winch of the ground station; Brüel & Kjær UA-1650 windscreen over the microphone to reduce wind sounds; Brüel & Kjær sound level meter 2250 | Three Brüel & Kjær 4189 microphones were positioned at equal distances along the driving route; The vehicle sounds were mainly emitted at the ground level and absorbed by padded microphone covers |

*Note*. VTOL: vertical take-off and landing. [a]The values refer to the ambient wind speed, but the towing speed was higher.

## 2.2 Listening experiment

### 2.2.1 Psychoacoustic Listening Laboratory

The listening experiment was conducted in the Psychoacoustic Listening Laboratory (PALILA) at the Faculty of Aerospace Engineering of Delft University of Technology. PALILA is a soundproof booth inside a separate room specifically designed to research the human perception of aeroacoustics sound sources, including aircraft, drones, and wind turbines. The booth is 2.32 m long, 2.32 m wide, and 2.04 m tall inside. The background noise level inside the room is 13.4 dBA. Merino-Martínez et al. (2023) describe the design and acoustic characterization in detail. PALILA's audio reproduction system is a Dell Latitude

7420 laptop (with an Intel® CoreTM i5-1145G7 vPro® processor and 16 GB of RAM) connected through a universal audio jack connector to a set of Sony WH-1000XM4 over-ear, closed-back headphones. The headphones allow for binaural hearing and have a 40 mm diameter dome-type driver unit, a frequency response between 4 Hz and 40 kHz, and a sensitivity of 105 dB/mW at 1 kHz. The audio reproduction system had been calibrated with a G.R.A.S. 45BB-14 KEMAR head and torso simulator. Participants are seated in the booth's center, and the laptop is placed on a table in front of them, as shown in Fig. 2.

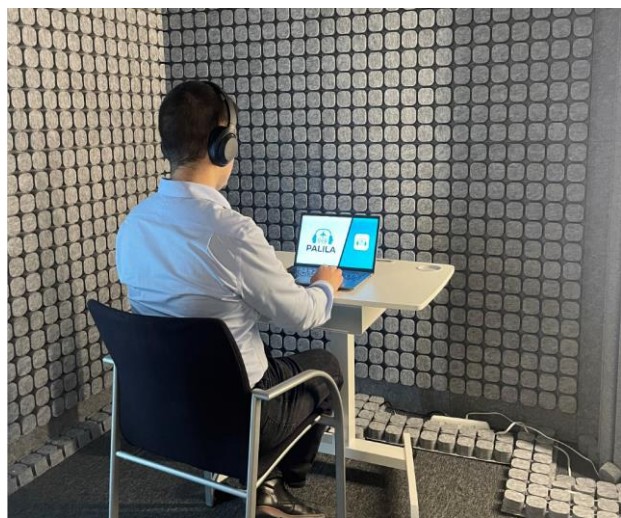

**Figure 2: Laboratory setup used for the listening experiment (source: authors' own).**

### 2.2.2 Participant recruitment and procedure

Participants were recruited using convenience and snowball sampling (Passer, 2014), mainly targeting students and employees. Participants were eligible to participate if they reported no hearing impairment and felt physically well on the day of the experiment. The study was conducted between June and September 2023. A trained experimenter instructed each participant individually, after which they completed the experiment independently.

In the first part of the questionnaire, participants were asked to self-report their hearing ability, hearing-affecting incidents (e.g., ear diseases, accidents, loud work environments), and well-being to establish their eligibility for participation. The second part of the questionnaire, the listening experiment, started with a practice round to get familiar with the process and the scales. It was followed by two counterbalanced blocks separated by an automatic and mandatory one-minute break: one block on AWES sounds and another on wind turbine sounds (the latter are not reported here). The sequence of the sound recordings within each block was randomized to minimize order and learning effects on participants' annoyance ratings (Passer, 2014). Participants listened to and evaluated one recording at a time. The recordings could not be replayed. The final part of the questionnaire asked about participants' noise sensitivity, familiarity with AWE, and demographic information. At

the end of the experiment, the experimenter debriefed the participant and handed over a 20-euro voucher as a participation reward. Participants took 22 minutes on average to complete the experiment, excluding the examiner's briefings.

### 2.2.3 Annoyance ratings and questionnaire

Noise annoyance was defined in accordance with the ISO 15666 standard as an individual's adverse reaction to noise, which may include feelings of dissatisfaction, bother, or disturbance caused by noise exposure (International Organization for Standardization, 2021). In line with the definition and recommended practice for psychoacoustic research (Alamir et al., 2019), annoyance levels were measured using the International Commission on Biological Effects of Noise (ICBEN) scale. For each sound recording, participants were asked to rate their experienced annoyance on both the verbal and numerical scale, and the average was calculated to increase measurement reliability (International Organization for Standardization, 2021). The 5-point verbal scale, ranging from "not at all" (0) to "extremely" (4), asked: "Imagine you are at home and hearing the noise at home; how much does the noise bother, disturb, or annoy you?" The 11-point numerical scale, ranging from 0 ("not at all") through 10 ("extremely"), asked: "Imagine you are at home and hearing the noise at home; what number from 0 to 10 best shows how much you are bothered, disturbed, or annoyed by the noise?". The wording of the scales was slightly adapted to acknowledge the laboratory setting.

To establish whether participants were eligible to partake in the study, their hearing ability was self-reported using a 5-point scale (from "poor" to "excellent"). Self-evaluations have been shown to provide a valid measure of individual hearing ability in the absence of audiometric testing (Hong et al., 2011). The occurrence of hearing-affecting conditions and incidents was also self-reported (e.g., hearing aid usage, ear diseases, accidents, tinnitus, loud work environments), and participants' well-being was queried (e.g., common cold, fatigue).

Noise sensitivity was assessed using the condensed 12-item version of the NoiSeQ scale. Participants rated their agreement with statements related to noise sensitivity in various contexts on a 4-point scale, ranging from "strongly disagree" (0) to "strongly agree" (3) (Griefahn, 2008). Sample items are "When I am at home, I quickly get used to noise" (reverse coded) and "When people around me are noisy, I find it hard to do my work". This scale has been shown to have high internal consistency ($\alpha = 0.87$) (ibid.).

Furthermore, whether participants were familiar with AWESs and had ever listened to one before was also assessed. Finally, the participants' age, gender, and education level were gathered. A graphical user interface (GUI) that guided participants through the entire questionnaire, including the listening experiment, was specifically developed for this experiment using MATLAB R2021b (see supplementary materials).

### 2.3 Participant characteristics

Of the 75 participants, 73.3% were male, 24% female, and 2.7% non-binary. The proportion of men was higher because participants were mainly recruited from a technical university. The age ranged from 18 to 66 years, with an average of 28 years

and a standard deviation of 9.57 years. The sample was overall highly educated, with 74.7% holding a Bachelor's or Master's degree, 16% currently or previously enrolled in university, and 8% having a doctoral degree. The average reported hearing ability was very good [Mean (M) = 4.07, standard deviation (SD) = 0.64, scale: 1-5], and the mean noise sensitivity was
195 medium (M = 1.56, SD = 0.38, scale: 0-3). About half of the participants reported being familiar with AWE (n = 37), but only 17.3% (n = 13) had listened to an AWES prior to the experiment. The high familiarity in the sample stems from the presence of a renowned research group on AWE at the faculty, exposing students and employees to the technology through institutional activities and research dissemination. However, this familiarity was largely theoretical, as most participants had not heard AWES sounds before the experiment. Therefore, the subsequent analyses did not consider experience with AWES sounds as
a confounding factor.

## 2.4 Post-processing of the results

### 2.4.1 Acoustic analyses

The EPNL metric (Kephalopoulos et al., 2014; Pieren et al., 2019) was used to explore how well conventional acoustic metrics can predict annoyance for AWESs. Furthermore, the following five SQMs (Merino-Martínez et al., 2021) were calculated for
each considered sound wave of every recording:

- Loudness (N): the perception of the sound magnitude corresponding to the overall sound intensity. Based on Zwicker's method, loudness was calculated using the ISO norm 532–1 (ISO/TC 43, 2017).
- Tonality (K): the perceived strength of unmasked tonal energy within a complex sound. Tonality was computed using
Aures' method (Aures, 1985).
- Sharpness (S): the high-frequency sound content. The DIN 45692:2009's (Deutsches Institut für Normung, 2009) method was used here.
- Roughness (R): the hearing sensation caused by modulation frequencies between 15 Hz and 300 Hz. Roughness was calculated according to Daniel and Weber (1997).
- Fluctuation strength (FS): assessment of slow fluctuations in loudness with modulation frequencies up to 20 Hz, with maximum sensitivity for modulation frequencies around 4 Hz. The method by Osses Vecchi et al. (2016) was used.

The five SQMs were evaluated over time using a subset of the full sound recordings to assess the repeatability of the metrics in the 25-second full-time span. To evaluate the sound quality through single quantities, the 5th percentile values were
220 used, which represent the level of each SQM exceeded during 5% of the total recording time (indicated henceforth by the subindex 5). From the SQMs, the PA metrics were calculated according to the models by Zwicker and Fastl (1999), More (2010), and Di et al. (2016). The general expression for the PA metric is

$$PA = N\left(1 + \sqrt{C_0 + C_1\omega_S^2 + C_2\omega_{FR}^2 + C_3\omega_T^2}\right), \tag{1}$$

where the term $\omega_S$ contains the sharpness S (and loudness N) contribution:

$$\omega_S = \begin{cases} 0.25(S - 1.75)log_{10}(N + 10), & \text{for } S \geq 1.75, \\ 0, & \text{for } S < 1.75. \end{cases} \tag{2}$$

The term $\omega_{FR}$ contains the contributions of the roughness R and fluctuation strength FS (and loudness N),

$$\omega_{FR} = \frac{2.18}{N^{0.4}}(0.4FS + 0.6R), \tag{3}$$

and the term $\omega_T$ contains the tonality K (and loudness N) contribution,

$$\omega_T \begin{cases} 0, & \text{for the model by Zwicker and Fastl (1999)} \\ (1 - e^{-0.29N})(1 - e^{-5.49K}) & \text{for the model by More (2010)} \\ \frac{6.41}{N^{0.52}}K, & \text{for the model by Di et al. (2016).} \end{cases} \tag{4}$$

Lastly, the coefficients $C_0$ to $C_3$ of Eq. (1) for each PA model are listed in Table 2. The conventional sound metrics, SQMs, and PA metrics were computed using the open-source MATLAB toolbox SQAT (Sound Quality Analysis Toolbox) v1.1 (Greco et al., 2023). Importantly, descriptive terms, such as "harsh," "beating," and "tonal," are later used to interpret the SQM results of the acoustic analysis. Participants did not provide these terms during the experiment.

**Table 2** Coefficients for Eq.(1) for each considered psychoacoustic (PA) model.

| PA model | $C_0$ | $C_1$ | $C_2$ | $C_3$ |
|---|---|---|---|---|
| Zwicker and Fastl (1999) | 0 | 1 | 1 | 0 |
| More (2010) | -0.16 | 11.48 | 0.84 | 1.25 |
| Di et al. (2016) | 0 | 1 | 1 | 1 |

**2.4.2 Annoyance ratings and percentage of highly annoyed respondents**

Following Brink and colleagues' approach (2016), verbal and numerical scale responses were linearly transformed to a 0-100 scale and averaged to obtain a total annoyance score per participant for each recording. The verbal and numerical scales were

strongly correlated in the present data, justifying calculating average scores (Tau-b item correlations were between 0.75 and 0.88). The average scores were used to determine the percentage of highly annoyed (%HA) participants for each recording. Following Miedema and Vos (1998) and the ISO standard (International Organization for Standardization, 2021), the top 28% of the scale were considered highly annoyed. That is, participants whose score was 72 or higher on the 100-point scale were classified as highly annoyed.

### 2.4.3 Linear-mixed effects models

Linear-mixed effects models were applied to identify significant predictors and to examine whether significant differences existed in the annoyance ratings across the three AWESs. Linear-mixed effects models can separate fixed effects (in this case, the acoustic predictors) from random effects (the participants with their individual characteristics). This type of hierarchical analysis has been successfully employed in past research on wind turbine noise annoyance (Merino-Martínez et al., 2021; Schäffer et al., 2016, 2019).

In this study, the sound recordings were nested within AWES types as each participant rated every recording that belonged to one of the three AWES types. Additionally, participants served as another level of nesting, as each participant contributed multiple ratings across the different AWESs. Following Judd et al. (2017), the nested structure was addressed by employing linear mixed-effects models with random effects for participants and AWES types. The conditions were contrast-coded to aid interpretation and included as random effects (ibid.). This approach allowed modeling the variability in annoyance ratings attributable to individual participants and differences between AWES types.

Following Aguinis and colleagues' step-wise approach (2013), participant characteristics were first included as fixed effects to determine their predictive value on annoyance ratings. Second, the SQMs were added as fixed effects, assessing each characteristic in separate models to avoid multicollinearity. Third, the impact of the SQMs was randomized to examine whether these effects varied between individuals. Fourth, interaction terms were included between participant characteristics and SQMs to explore whether the participant characteristics could explain individual differences in the impact of SQMs on annoyance ratings.

Finally, using the -2-log likelihood ratio, the goodness-of-fit for the final linear mixed-effects models was assessed to quantify the variance explained by the fixed factors alone and by both fixed and random factors. Separate linear-mixed effects models, including EPNL or the PA models as predictors, evaluated how effectively these (psycho)acoustic metrics could predict the annoyance ratings. All statistical analyses were performed using the software R version 4.4.0 (R Core Team, 2023), and linear mixed-effects models were fitted using the 'lme4' package (Bates et al., 2024).

## 3. Results

### 3.1 Acoustic results

The time-frequency sound levels were represented as spectrograms (see Fig. 3). The spectrograms were calculated with a sampling frequency of 48000 Hz for every audio sample using 4800 samples per time block (i.e., 0.1s) with Hanning windowing and 50% data overlap. These parameters provided a frequency resolution ($\Delta$f) of 10 Hz.

      For AWES A, the lower frequencies (0-1 kHz) exhibited higher sound levels, which decreased as the frequency increased. The spectrograms confirmed that the recordings were representative. For AWES B, the highest sound levels were

found at extremely low frequencies, up to approximately 200 Hz. Sound levels decreased between 200 Hz and 1 kHz but increased again in the frequency range between 1 and 3 kHz. AWES B exhibited a periodic sound pattern over time, likely due to its circular flight trajectory. A periodic sound pattern was observed for recordings corresponding to AWES C, characterized by a significant absence of sound levels in the frequency range between 200 Hz and 1.2 kHz (C1 and C2) and between 200 Hz and 2 kHz (C3). These periodic behaviors are again attributed to the circular flight trajectory. For C1, the acoustic energy was

predominantly concentrated between 15-25 s and in the frequency range between 1.2 and 4 kHz. C2 showed consistent sound levels, peaking between 1.2 and 5 kHz. Conversely, C3 displayed higher levels within the first 8 s at 2-5 kHz.

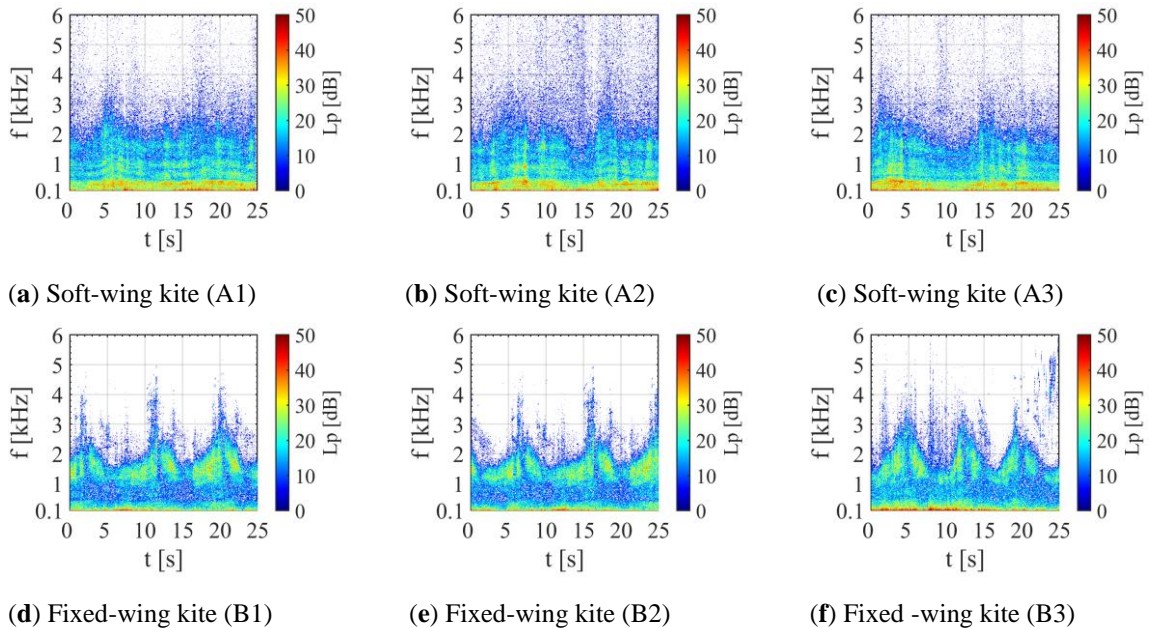

(**a**) Soft-wing kite (A1)       (**b**) Soft-wing kite (A2)       (**c**) Soft-wing kite (A3)

(**d**) Fixed-wing kite (B1)       (**e**) Fixed-wing kite (B2)       (**f**) Fixed -wing kite (B3)

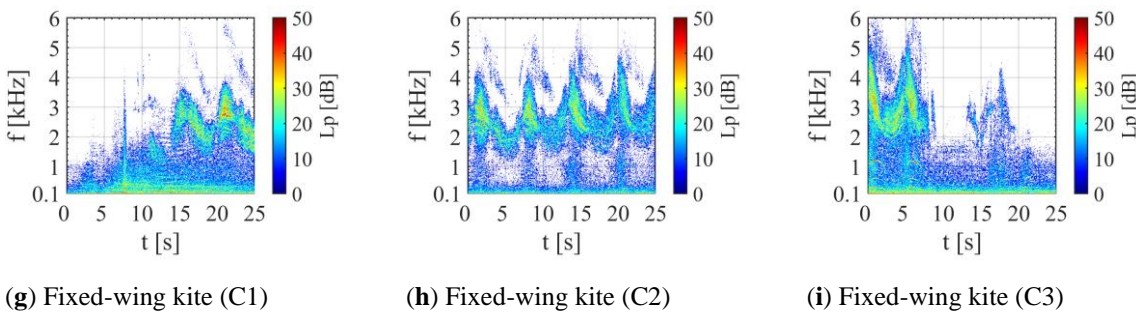

(**g**) Fixed-wing kite (C1)    (**h**) Fixed-wing kite (C2)    (**i**) Fixed-wing kite (C3)

**Figure 3: Spectrograms corresponding to each recording.**

Time-averaged sound pressure levels (SPL) were computed, as shown in Fig. 4, to compare the sound levels produced by each type of AWES. For AWES A, AWES B, and the second recording of AWES C (C2), SPLs were averaged over the whole 25 s recording duration. In contrast, for the first and third recordings of AWES C (i.e., C1 and C3), the averages considered the last 10 s and first 8 s, respectively, when the kite noise was perceivable. In the listening experiment, the full recordings were used. Only slight variations were observed when considering the entire recording.

SPLs were virtually the same across the entire frequency range for AWES A, displaying a bump in the 200 Hz to 2 kHz range. AWES B showed similar trends and sound levels across the recordings, although there was a difference of approximately 4 dB between B1, B2, and B3 for frequencies up to 1 kHz. For frequencies higher than 1 kHz, the SPLs were nearly identical across all B-recordings. Regarding AWES C, time-averaged SPLs showed more significant differences for frequencies below 1.6 kHz, with C3 having the highest sound levels, followed by C1 and C2. Additionally, C3 elucidates peaks

that suggest tonal behavior within this frequency range. On the other hand, the frequencies above 1.6 kHz were similar among the recordings, though C3 exhibited higher sound levels than C1 and C2 in the 3 to 5 kHz range. It was also observed that AWES A and AWES B had higher sound levels than AWES C, particularly for frequencies below 100 Hz, see Fig. 5. The SPLs in C1 and C3 exhibited a tonal behavior in the frequency range of 60 to 1300 Hz, which is believed to be related to the ram-air turbine. The flight patterns for both AWESs B and C are circular, which may induce specific turbulent flow

characteristics around the kite's surfaces and structures for frequencies higher than 1 kHz. In contrast, AWES A, which follows a figure eight flight pattern, did not show this acoustic behavior.

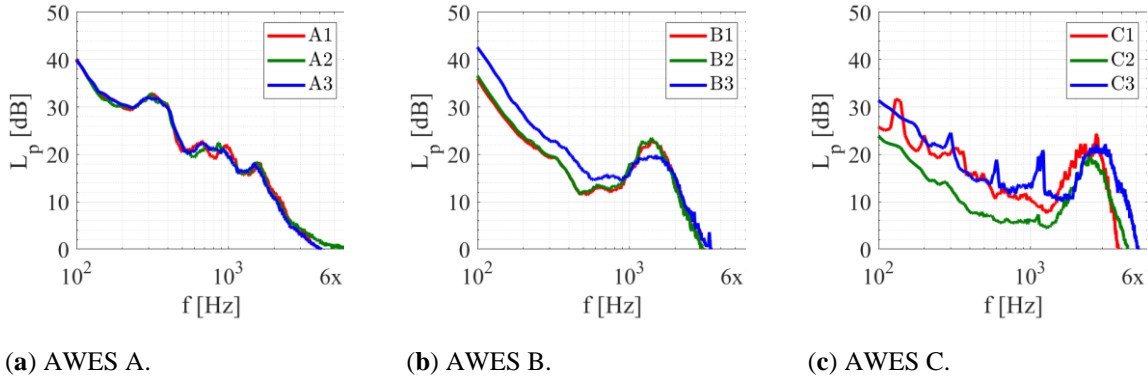

(**a**) AWES A.        (**b**) AWES B.        (**c**) AWES C.

**Figure 4: Time-averaged sound pressure levels for each airborne wind energy system (AWES).**

Fig. 5 shows the comparison of time-averaged SPLs between each AWES. For this purpose, one representative case of each AWES configuration (i.e., A2, B3, and C3) has been selected. The soft-wing (A2) and fixed-wing (B3 and C3) kites exhibited a broadband acoustic trait. However, the fixed-wing kites showed an acoustic bump at high frequencies (950 to 3420 Hz for B3 and 1910 to 5180 Hz for C3) that the spectrum of the soft-wing kite did not. Additionally, the spectrum of C3 revealed narrowband peaks around 300 Hz, 600 Hz, and 1200 Hz, which could be related to the ram-air turbine. The 600 Hz

and 1200 Hz peaks also seem to be harmonics of the rotations of the ram-air turbine (300 Hz), as they were equally spaced.

   The broadband acoustic nature of the soft-wing kite is believed to arise from its flexible, deformable structure and complex, turbulent aerodynamic interactions. This acoustic component was also higher than the broadband acoustic signature found in fixed-wing kites (i.e., 180 to 1000 Hz for B3 and 100 to 1900 Hz for C3). This may be related to the soft-wing kites' fabric-based material, which promotes constant deformation and fluttering, creating turbulence that produces a stronger

broadband noise component than fixed-wing kites. This turbulence-induced acoustic trait was spread over a broad frequency range, contributing to the broadband nature of the noise. For B3, the noise bump in the 1-2 kHz range might be due to vortex-shedding frequencies around the kite's body or edges. For C3, the additional components introduced by the ram-air turbine could shift these frequencies upwards to the 2-3 kHz range.

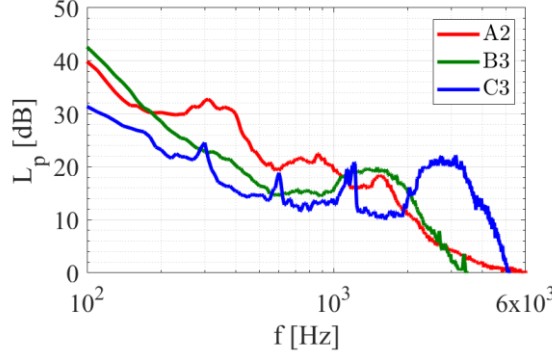

**Figure 5: Comparison of time-averaged sound pressure levels of one representative recording for each airborne wind energy system (AWES).**

### 3.2 Psychoacoustic sound quality metrics of AWESs and their relation to annoyance

An analysis of the SQMs (Table 3) revealed differences across the three AWESs, as illustrated in the violin plots in Fig. 6. Regarding loudness (Fig. 6a), AWES A recordings exhibited nearly identical values, consistent with the spectra shown in Fig. 4a. AWES B recordings showed slight variations, with B3 displaying higher loudness levels than B1 and B2. This difference can be attributed to higher noise levels in the 100 to 1000 Hz range for B3 (Fig. 4b), likely due to its closer proximity to the microphone. For AWES C, C3 exhibited higher loudness values than C1, and C1 had higher values than C2. This pattern aligns with the spectra depicted in Fig. 4c.

Although the spectra for C1 and C3 appear similar, a noticeable difference in sound pressure levels (Lp) is observed for frequencies below 1.6 kHz, particularly for C2, which exhibits lower Lp compared to C1 and C3. Additionally, C3 elucidates peaks that suggest tonal behavior within this frequency range. Human hearing is most sensitive to frequencies between 2 and 5 kHz, as illustrated by the equal-loudness contours (ISO 226). Observing Figure 4c, it can be noted that C3 displays higher Lp values in the frequency range of 3–5 kHz. Among all the recordings, C3 showed the highest 5% percentile loudness values, potentially related to the sudden increase in sound levels around 1200 Hz. This sudden sound increase could be attributed to the vibration of the rigid structure of the fixed-wing kite compared to the soft-wing kite (i.e., inflatable kite made from fabric) or to the ram-air turbine on the fixed-wing kite.

Regarding tonality (Fig. 6b), both AWES A and AWES B showed relatively low values compared to AWES C. This behavior can be explained by the narrowband peaks in the sound spectra observed in C1 and C3, as shown in Fig. 4c. The soft-wing kite generally exhibited the lowest tonality values, which can be explained by its tendency to produce more broadband and less tonal sound. Most noise from soft-wing kites is due to fabric flutter and aerodynamic noise.

Regarding sharpness (Fig. 6c), AWES C notably showed higher values than AWES A and B, consistent with the sound spectra (Fig. 4) since the sharpness calculation emphasizes frequencies for critical bands above 15 Bark (corresponding to approximately f = 2700 Hz). Additionally, C3 presented the sharpest sound, which aligns with the definition of sharpness since this kite reported higher sound values than the other kites for frequencies above 2700 Hz (Fig. 5). Roughness (Fig. 6d) and fluctuation strength (Fig. 6e) quantify the perception of modulated sounds with a modulation frequency between 15 Hz and 300 Hz and below 20 Hz, respectively.

Regarding roughness, B3 was perceived as the 'harshest' compared to all other recordings, while the AWES C recordings were the 'least harsh.' Regarding fluctuation strength, AWES B was observed to have the 'strongest beating' effect, whereas AWES C was 'less pulsating.'

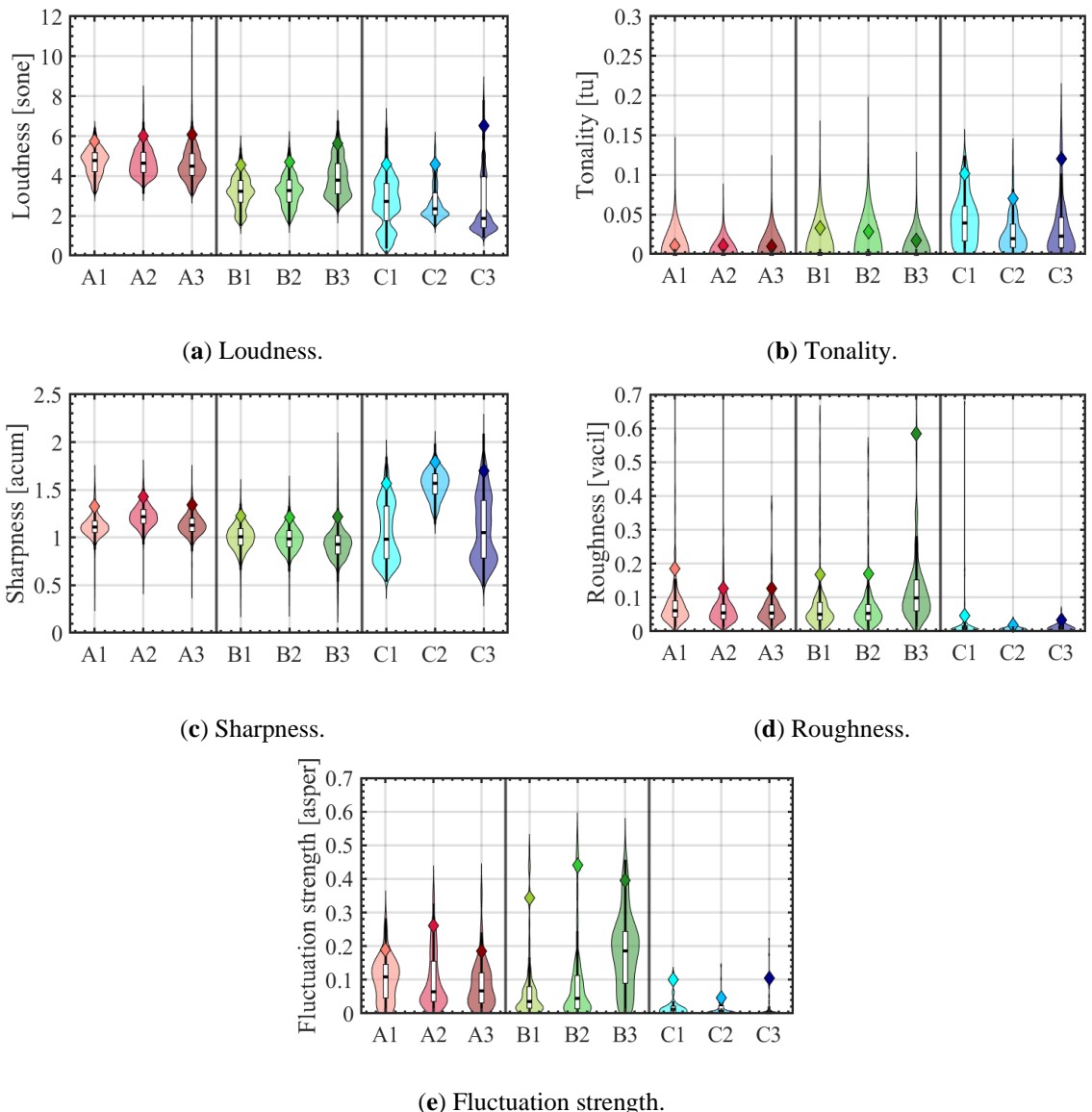

(**a**) Loudness.

(**b**) Tonality.

(**c**) Sharpness.

(**d**) Roughness.

(**e**) Fluctuation strength.

**Figure 6: Violin plots of sound quality metrics for all recordings. Plot widths represent the probability density at given values in the y-axis. Diamond markers indicate the 5th percentile values (i.e., the values exceeded 5% of the signal time, as explained in section 2.4.1). In each boxplot, the central horizontal line denotes the median values, and the edges of the white box plot represent the 25th and the 75th percentiles.**


**Table 3** The 5th percentile values of the five sound quality metrics per recording.

| Recording | L5 (sone) | K5 (tu) | S5 (acum) | R5 (vacil) | FS5 (asper) |
|---|---|---|---|---|---|
| A1 | 5.75 | 0.011 | 1.32 | 0.18 | 0.19 |
| A2 | 5.99 | 0.011 | 1.42 | 0.13 | 0.26 |
| A3 | 6.90 | 0.010 | 1.34 | 0.13 | 0.19 |
| B1 | 4.55 | 0.033 | 1.22 | 0.17 | 0.34 |
| B2 | 4.71 | 0.028 | 1.21 | 0.17 | 0.44 |
| B3 | 5.62 | 0.018 | 1.22 | 0.58 | 0.39 |
| C1 | 4.57 | 0.102 | 1.57 | 0.05 | 0.10 |
| C2 | 4.57 | 0.071 | 1.79 | 0.02 | 0.05 |
| C3 | 6.53 | 0.121 | 1.70 | 0.03 | 0.10 |

### 3.2.1 Analysis of annoyance ratings

The mean annoyance ratings for the different AWES types ranged from approximately 34 for AWES A to 54 for AWES C (Fig. 7). In comparison, Merino-Martínez et al. (2021) reported average annoyance ratings of about 61-72 (converted from the ICBEN 11-point scale to a 0-100 scale) for wind turbine sound in a laboratory experiment. However, the $L$p,A,eq values in their study were lower, at around 38 dBA, than those used here.

The percentage of highly annoyed participants (%HA) per AWES type varied between approximately 1% and 19% (Table 4), with AWES C showing the highest %HA, followed by AWES B and then A. This trend aligns with the previously reported higher tonality and sharpness values for AWES C compared to B and A. The observed %HA range is slightly narrower than the 2 to 34% predicted by Schäffer et al. (2016) for wind turbine sound exposure in laboratory settings with an $L$p,A,eq range of 35 to 45 dBA.

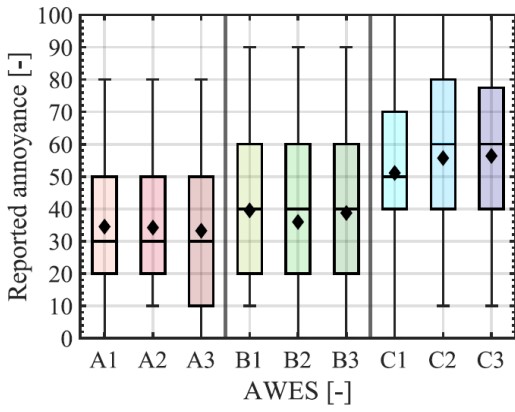


Figure 7. Distribution of annoyance ratings per recording. In each boxplot, the diamond marker denotes the mean value; the central horizontal line denotes the median values; the edges of the box are the 25th and the 75th percentiles; and the whiskers extend to the most extreme data points.

**Table 4** Percentage and frequency of highly annoyed participants (%HA) per airborne wind energy system (AWES).

| AWES | %HA |
| --- | --- |
| A (soft-wing) | 1.3 (2) |
| B (fixed-wing) | 6.7 (5) |
| C (fixed-wing) | 22.7 (17) |

Pairwise comparisons between AWESs were conducted using the linear mixed-effects model. The model revealed significant differences across all three AWESs (all p-values < 0.05). In line with the previous results on the percentage of highly annoyed participants, fixed-wing kite C was, on average, rated as the most annoying [Mean (M) = 54.39, standard
deviation (SD) = 22.91], followed by fixed-wing kite B (M = 39.78, SD = 22.04) and soft-wing kite A (M = 33.98, SD = 20.47).

A separate linear mixed-effects model was calculated to examine whether noise annoyance depended on participant characteristics. Noise sensitivity was significantly related to annoyance [t-statistic (t) = 2.035, p < 0.050], indicating that individuals more sensitive to noise generally rated the recordings as more annoying. Age (t = 1.332, p = 0.187) and familiarity
with AWE (t = 0.056, p = 0.956) were not significantly related to annoyance ratings.

A linear mixed-effects model of the relation between annoyance ratings and SQMs showed that sharpness significantly predicted annoyance (t = 2.285, p = 0.023), while tonality (t = 0.933, p = 0.393), loudness (t = 0.416, p = 0.695), roughness (t = -0.601, p = 0.574), and fluctuation strength (t = 0.676, p = 0.529) did not. Fig. 8 displays a significant and strong

relationship between sharpness and annoyance (r = 0.863, p = 0.002). The results align with the finding that the annoyance ratings were significantly higher for AWES C, which exhibited higher sharpness values than AWESs A and B.

To evaluate whether the impact of the SQMs on annoyance ratings varied across participants, models incorporating SQMs as fixed effects were compared with those treating them as random effects, computing the -2-log likelihood ratio between these models. The models treating all SQMs except fluctuation strength as random effects—loudness ($\chi^2(1) = 18.725$, $p < 0.001$), sharpness ($\chi^2(1) = 9.121$, $p = 0.003$), tonality ($\chi^2(1) = 7.146$, $p = 0.008$), and roughness ($\chi^2(1) = 8.723$, $p = 0.003$)— showed a significantly improved fit compared to the models treating them as fixed effects. This suggests that all tested SQMs, except for fluctuation strength, influenced annoyance ratings differently across individuals. These variations may reflect individual differences in factors such as noise sensitivity, age, or familiarity with AWES, which can shape how participants perceive and react to specific sound qualities.

To explore whether these individual characteristics could account for the observed differences, interaction effects between the SQMs and participant characteristics (i.e., age, AWE familiarity, and noise sensitivity) were included in the models with the random SQM effects. The results revealed that the interaction effect of participant characteristics and loudness was significant for AWE familiarity (t = -2.902, p = 0.005) but not for age (t = 0.988, p = 0.327) nor noise sensitivity (t = 0.699, p = 0.049). That is, the effect of loudness on annoyance was weaker for those more familiar with AWE. This familiarity may be intertwined with more positive attitudes toward AWE, potentially explaining the lower levels of noise annoyance observed—a pattern reported in studies on wind turbines (Dällenbach and Wüstenhagen, 2022; Hoen et al., 2019; Hübner et al., 2019). Furthermore, the interaction effect of participant characteristics and tonality was significant for age (t = -2.233, p = 0.028) but not for AWE familiarity (t = -0.452, p = 0.652) nor noise sensitivity (t = 0.045, p = 0.964). This suggests that the effect of tonality on annoyance was weaker for older individuals, also independent of participants' self-reported hearing ability.

The interaction effects of participant characteristics and sharpness, roughness, and fluctuation strength were not significant for any of the included participant characteristics (with p-values ranging from 0.139 to 0.915). The full model, including all interactions between participant characteristics and SQMs, explained 19% of the variance in annoyance scores due to the fixed effects alone and 82% of the variance when both fixed and random effects were considered.

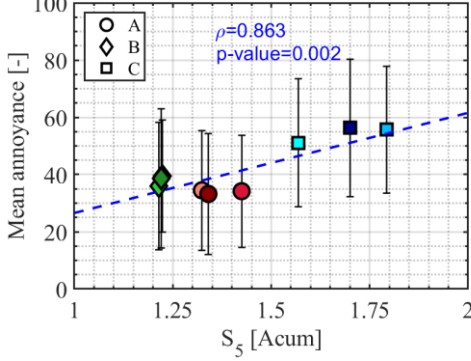

 **Figure 8: Linear correlation between the average annoyance rating per recording and sharpness across the airborne wind energy systems (AWESs).**

### 3.2.2 Validity of conventional and psychoacoustic metrics in predicting annoyance ratings for AWESs

It was explored with linear mixed-effects models to what extent EPNL as a conventional metric and the Psychoacoustic Annoyance (PA) (i.e., Zwicker and Fastl, 1999; More, 2010; Di et al., 2016) models predict the annoyance ratings reported in the experiment. Table 5 presents the values used to perform these analyses. EPNL ($t = 0.700$, $p = 0.515$) did not significantly predict the annoyance ratings. Linear mixed-effects models comparing the annoyance ratings with the estimated annoyance scores (5th percentile values) for each PA metric separately (Zwicker and Fastl, 1999; More, 2010; Di et al., 2016) showed that the PA metrics did not significantly predict the annoyance ratings: Zwicker and Fastl ($t = 0.117$, $p = 0.911$), More ($t = 0.541$, $p = 0.612$), and Di et al. ($t = 0.466$, $p = 0.661$). Because PA metrics heavily depend on loudness, the aforementioned normalization of all recordings might explain why the PA metrics were not significant predictors.

**Table 5** EPNL and 5th percentile values of the three psychoacoustic (PA) models per recording.

| | Conventional metric | PA models | | |
|---|---|---|---|---|
| Recording | EPNL (EPNLdB) | Zwicker and Fastl | More | Di et al. |
| A1 | 61.2146 | 6.6344 | 6.2142 | 6.6352 |
| A2 | 60.3708 | 6.6663 | 6.3786 | 6.6832 |
| A3 | 59.7796 | 6.9043 | 6.5423 | 6.9138 |
| B1 | 59.2138 | 5.1527 | 4.9296 | 5.2142 |
| B2 | 58.8830 | 5.2055 | 4.6478 | 5.2324 |
| B3 | 60.8026 | 7.5308 | 6.5923 | 7.5316 |
| C1 | 61.6614 | 4.8561 | 4.9263 | 5.3365 |
| C2 | 61.9017 | 4.7786 | 4.8967 | 5.1501 |

| C3 | 63.7321 | 7.0838 | 7.4605 | 8.2037 |
|---|---|---|---|---|

## 4 Discussion

Through a controlled listening experiment, this study explored the relationship between sound quality metrics (SQMs) and noise annoyance for airborne wind energy systems (AWESs). Sharpness emerged as the sole SQM that significantly predicted annoyance. Fixed-wing systems were perceived as more annoying than the soft-wing kite, likely due to their sharper and more tonal sound profiles. The higher loudness values found for the soft-wing kite can be explained by its aerodynamic characteristics that produce more broadband and less tonal sound. In contrast, the higher tonal sound signature of fixed-wing kite C is attributable to its ram-air turbine. Participant characteristics moderated the effects of certain SQMs: Participants familiar with AWESs were less annoyed by louder recordings than unfamiliar participants, and older individuals were less annoyed by more tonal sounds than younger individuals. These moderation effects should be cautiously interpreted because they could be random due to the non-probability sampling and the lack of representativeness of the sample in this study. Contrary to prior research on wind turbines and drones (Kawai et al., 2024; Merino-Martínez et al., 2021), conventional noise metrics like the effective perceived noise level (EPNL) and Psychoacoustic Annoyance (PA) models did not predict annoyance effectively, likely due to the normalization of sound pressure levels across recordings in this study.

This study builds on findings from related research on wind turbines and drones, which share acoustic and operational parallels with AWESs. Research on wind turbines often focuses on sound pressure levels, while studies on SQMs remain scarce. However, some work suggests that tonality and loudness predict annoyance (Merino-Martínez et al., 2021; Yonemura et al., 2021). Drone research, by contrast, has more thoroughly explored SQMs, with loudness, tonality, and sharpness consistently identified as key predictors of annoyance (Casagrande Hirono et al., 2024; Green et al., 2024; Kawai et al., 2024; König et al., 2024; Torija and Nicholls, 2022). These systems are operationally similar to AWESs, particularly in their dynamic flight stages and use of propeller-like mechanisms. For drones, annoyance peaks during take-offs and landings, which could also be the case for AWESs and should be investigated in future studies. While the current study confirmed sharpness as a critical predictor of annoyance, tonality, and loudness were significant only in interaction with participant characteristics (i.e., age and familiarity, respectively).

The findings should be interpreted in light of several limitations. First, the study used a convenience sample, primarily recruiting students and employees from a technical university. This introduces potential selection bias and limits the generalizability of the results to the broader population, especially residents in areas where AWESs might be deployed. The sample was rather young, predominantly male, and highly educated, which may not accurately represent the diversity of individuals who might encounter AWES noise in real-world settings. Second, the controlled laboratory setting ensured consistency but did not replicate real-world listening conditions. Participants rated annoyance without contextual factors like

visual exposure to AWESs, other environmental sounds, or social and psychological influences (e.g., fairness perception of the planning process) that typically influence noise annoyance in the field. Third, participants' short-term annoyance ratings do not capture the potential cumulative effects of prolonged or repeated exposure. Fourth, although 75 participants are on the high end of sample sizes used in listening studies (Alamir et al., 2019), the statistical power to detect subtle effects or interactions, particularly those involving individual differences (e.g., age, familiarity, noise sensitivity) was limited. Fifth, the study investigated only three AWES prototypes (one soft-wing and two fixed-wing systems). The results may not generalize to other AWES designs or operational configurations. Sixth, the study's methodology faced several challenges related to sound recordings, particularly concerning the varying distances to the microphone (100–700 m) and the moving nature of the kites. While normalization to 45 dBA mitigated some inconsistencies, the dynamic sound signatures created by the kites' flight patterns introduced additional variability compared to the noise emissions of stationary wind turbines. Additionally, the location of the observer or microphone significantly influences noise perception because the acoustic prominence of different system components varies depending on the vantage point. For example, certain components, such as the kite, may dominate acoustically when the observer is positioned directly below or in line with the kite's trajectory. In contrast, noise from the generator or tether vibrations may become more prominent at close distances to the ground station. Furthermore, environmental factors such as wind noise and ground reflections may have influenced the recordings despite mitigation efforts using windscreens. These limitations underscore that the results have only restricted applicability to the field. Schäffer et al. (2016) highlighted these challenges, emphasizing that laboratory and field studies should be viewed as complementary rather than directly comparable.

To address these limitations and advance the understanding of AWES acoustics, future research should explore annoyance during different phases of the AWES pumping cycle to identify the stages that cause the most impact and guide the development of targeted mitigation strategies. Research should also be expanded to include a wider variety of AWES prototypes, capturing these systems' diverse noise profiles and operational characteristics. Conducting field studies that account for environmental and contextual factors, such as background noise, visual exposure, and long-term sound patterns, would provide more ecologically valid insights into real-world annoyance. Additionally, studies should examine the effects of extended exposure and repeated noise events on annoyance, focusing on potential consequences like sleep disturbances and stress. It is equally important to engage a broader range of demographic groups, particularly those living near current or proposed AWES installations, to ensure that findings are representative of affected populations. Finally, developing noise prediction models specifically tailored to AWES should be prioritized. These models should incorporate the dynamic operational characteristics of AWESs, such as variations in speed and trajectory, to improve their accuracy and relevance for mitigating noise annoyance.

The design flexibility of AWESs provides unique opportunities to mitigate annoyance through targeted optimizations:

- Tunable system parameters: Unlike wind turbines, AWESs allow adjustments to size, speed, altitude, and flight patterns. For example, larger kites flying higher could reduce sharpness and modulation effects, while faster, lower-altitude configurations might be suitable where loudness is less critical.

- Ram-air turbine optimization: The onboard ram-air turbine supplies power to the kite control unit and sensors and can be designed for minimal noise emissions without significantly affecting the system's energy output or economic performance.
- Flight path design: Optimizing flight paths, such as larger figure-eight loops, could reduce modulation effects while adjusting reel-in and reel-out speeds may help minimize tonal noise.
- Customized configurations: AWESs can be tailored to site-specific conditions, balancing energy output with acoustic considerations. For example, quieter configurations may be prioritized in residential areas, while efficiency-driven designs might be more suitable for remote locations.
- Proactive engagement: Industry stakeholders should involve communities early in the planning process, using psychoacoustic data to communicate potential impacts and suggest mitigation strategies transparently.

By leveraging these design possibilities, the AWE industry could effectively address noise concerns, promoting broader technology acceptance.

## 5 Conclusion

This study identified sharpness as a key predictor of noise annoyance for AWESs, with fixed-wing kites eliciting higher annoyance than soft-wing designs. Fixed-wing kites had sharper and more tonal sound profiles, while the soft-wing kite had higher loudness values. Participant characteristics influenced the impact of loudness and tonality on annoyance, highlighting the complexity of subjective noise perception. The findings further emphasize the limitations of conventional noise metrics in assessing AWES noise, suggesting the need for tailored acoustic models. The industry can address noise challenges by integrating psychoacoustic considerations into the design and operation of AWESs, such as optimizing system parameters and flight patterns. Future research should expand on these findings by incorporating field studies, long-term exposure assessments, and analyses of diverse prototypes.

**Data availability**

The data that support the findings of this study are openly available in 4TU.ResearchData, reference number [doi: 10.4121/2716b49f-b44c-400a-a873-eea276b081f6].

**Author contributions**

HS, RMYV, DR, RMM, and RS planned the campaign; HS performed the measurements; HS, RMYV, and PvG analyzed the data; HS and RMYV wrote the manuscript draft; DR, RMM, PvG, and RS reviewed and edited the manuscript.

**Competing interests**

At least one of the authors is a member of the editorial board of Wind Energy Science. Roland Schmehl is a co-founder of and advisor for the start-up company Kitepower B.V.

**Acknowledgments**

We thank the three airborne wind energy developers and Naomi Bouman for providing the sound measurements and acoustic data. We are grateful to Niels Adema for helping with planning the campaign and supporting the measurements. We would also like to thank the anonymous reviewers for their valuable feedback and constructive comments, which have greatly improved the quality and clarity of this manuscript.

**Financial support**

This work was supported by the Dutch Research Council (NWO) and Kitepower B.V. [grant number 17628]. The research work of Roberto Merino-Martínez is funded by the project Listen to the future [project number 20247] of the research program Veni 2022 (Domain Applied and Engineering Sciences), which NWO partly finances.

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
