# Peer review of "Exploring noise annoyance and sound quality for airborne wind energy systems: Insights from a listening experiment"

_Wind Energy Science, 2024_

## Author Response (AR1)

| Reviewer 1 | Response to reviewer 1 | Changes to Manuscript |
|---|---|---|
| 1) General comment

The present study is interesting and innovative as it gives insight into the properties of a new and developing technology. It applies knowledge about a more established form of wind energy, namely that sound emissions are a central issue for residents, to a context where conflictual cases can yet be prevented. Analyzing the psychoacoustic metrics of different form factors is important in order to understand how design choices affect the experience.

The article is very well written, methodology and results are presented clearly. | We thank you for your kind words. | No changes were necessary in response to this comment. |

| 2) General comment continued. | Thank you for highlighting this important point. We have expanded the Introduction and Discussion to include comparisons with (psycho)acoustic studies on similar technologies, such as wind turbines and drones. These comparisons help contextualize the findings and demonstrate whether the observed patterns in sound quality metrics (SQMs) and annoyance ratings are consistent or unexpected. Additionally, we now address the design implications of our results. | The Introduction and Discussion sections were expanded to compare the findings with existing (psycho)acoustic studies on similar technologies. We also included implications of the results for AWES design improvements. |
| --- | --- | --- |
| Where it falls short of its potential yet is in the discussion of what the study means for AWE. Both in the introduction as well as when discussing the results there is little reference to other studies with similar approaches. As there is not a lot of research on AWE specifically, a comparison to the psychoacoustic metrics of other, to some degree similar technologies would help a reader gauge the significance of the results. As it is, a reader without specific psychoacoustic background knowledge would have a difficult time judging whether the results are surprising or expected. Furthermore, it could be clarified what the results imply for the design of AWESs. Do they suggest improvements to be made, for example? | | |

| **SPECIFIC COMMENTS:** | | |
|---|---|---|
| 3) Section 2.1: How comparable are the three recordings per AWES to each other? Are the recordings selected to cover varying situations or are they supposed to be consistent? The spectrograms in section 3.1 reveal remarkable differences for kite C, whereas the recordings of the other two kites seem fairly homogenous. | The three recordings per AWES were selected to represent typical sound emissions during the reel-out phase of the respective system. Given that kite C was towed by a truck, there is larger variability within the 3 recordings in comparison to kites A and B. | We have added this explanation in Section 2.1 "Sound recordings" to clarify the selection process and variability of the recordings. |
| 4) 84: Please add some explanation why only recordings of the reel-out phase were chosen. Are there differing acoustic properties of the phases? Is the reel-out phase more relevant? Are they similar enough that the results of one should translate to the other? | The reel-out phase was chosen because it is acoustically the most dominant and operationally consistent phase across all AWESs. During this phase, the kite operates in crosswind maneuvers at high flight speeds, while the reel-out speed is kept relatively low to maximize the energy production period. This operational setup implies that sound emissions from the kite, including contributions from the onboard ram-air turbine, wing flutter, and tether vibrations, are the most significant. In contrast, sound emissions from the ground station (e.g., the generator) are comparatively minor due to the low reeling speed. Although other phases, such as reel-in and hovering, also produce sound emissions, these are typically less prominent | We have clarified this point in Section 2.1 to justify the focus on the reel-out phase. |

| | | |
|---|---|---|
| | and exhibit greater variability due to differences in operational settings. By focusing on the reel-out phase, the study ensured that the results reflected a consistent and acoustically relevant scenario, providing a reliable basis for assessing noise  annoyance. | |
| 5) 256-257: Please elaborate on why the circular flight pattern results in a bump above 1 kHz. How does this explanation relate to the one from l.274-275, stating the bump is a result of rigid materials? | We appreciate the opportunity to clarify this point. We now specified in the results that the circular flight patterns of AWESs B and C could cause the bump sound pattern, possibly due to induced specific turbulent flow characteristics around the kite's surfaces and structures for frequencies higher than 1 kHz. Additionally, at the end of the paragraph of Figure 5, we mentioned that for B3, the noise bump in the 1-2 kHz range might be due to vortex shedding frequencies around the kite's body or edges and that  for C3, the additional components introduced by the turbine could shift these frequencies upwards to the 2-3 kHz range. | Explanations were added in the Results section and accompanying Figure 5 to clarify the relationships between circular flight patterns, vortex shedding, and the observed bumps in sound frequencies. |
| 6) Section 3.2: Is there a criterium applied to judge which differences are meaningful? Regarding the loudness of kite C, for example, the text says that C3 exhibits higher values than C1. Looking at Figure 6 this is true regarding | Thank you for raising this point. In psychoacoustic studies, differences in individual SQMs are considered meaningful if they are larger than a threshold value called Just Noticeable Difference (JND). These threshold values can be found in the literature, e.g., Osses et al., 2023. We computed the JND for A1, B1 and | The relevant bar plots can be included in the supplements if requested. |

the maximum values, though not with respect to the distribution or median values. If the statement is based on the 5th percentile what is the threshold of a meaningful difference?

C1 as references and compared them to the values from the literature. The bar plots below display the JND per metric to show what deltas are significant (i.e. perceived as different). If desired, we can include the plots in the supplements.

[1] Osses, A., Greco, G.F., and Merino-Martinez, R., "Considerations for the perceptual evaluation of steady-state and time-varying sounds using

psychoacoustic metrics," 10th Convention of the European Acoustics Association (Forum Acusticum), 11-15 September 2023, Torino, Italy.

| | | |
|---|---|---|
| 7) 329: Please add some words of guidance to Figure 8, help the reader interpret how strong the effect is. | We appreciate the feedback. | Additional guidance was added to the Results to help readers interpret the strength of the effect. |
| 8) It was not the aim of this study to compare annoyance levels of AWESs to other emission sources. But can you provide some point of comparison to understand whether the annoyance levels are particularly high or low? At least, it might be worth giving a verbal orientation on how to interpret the annoyance values within the scale (l.314-315). | Thank you for this suggestion. While direct comparisons to other noise sources were beyond the scope of this study, we agree that providing context is helpful. We have added comparisons with Merino-Martínez et al. (2021) and Schäffer et al. (2016), who assessed annoyance for wind turbine sound in laboratory settings under similar conditions. Unfortunately, we couldn't find more relevant studies because existing research on wind turbine noise annoyance varies significantly in methodology, including differences in annoyance scales, thresholds for defining "highly annoyed," sound pressure levels, and study settings (laboratory vs. field). | Comparisons with existing research on wind turbine noise annoyance were added to provide context and help readers interpret the annoyance ratings. |

| | | |
|---|---|---|
| 9) You state that the moderation effects might be tied to the sample. There are more specific explanations possible. Familiarity with AWE at a technical university with ties to AWE development might imply a more positive attitude, hence affecting annoyance ratings. On the other hand, the sample is particularly young, raising questions about the effect of age on tonality. Given the age distribution, what age differences result in the effect of age on tonality? | We agree with you that participants' familiarity with AWESs may reflect a more positive attitude, leading to lower annoyance ratings. We have added the following: "This familiarity may be intertwined with more positive attitudes toward AWE, potentially explaining the lower levels of noise annoyance observed—a pattern consistently reported in studies on wind turbines."

Regarding age, we acknowledge that the sample is relatively young, which may limit our ability to generalize the age-related effects on tonality. That is why we mention in the Discussion that the moderation effects should be viewed cautiously due to the non-probability sampling used and the sample's lacking representativeness. Future studies with a more balanced age distribution would help clarify this relationship. | We have added text to the Results section discussing the potential influence of participant familiarity and attitudes. |
| **TECHNICAL CORRECTIONS:**

10) 14 & 75: Mostly, in this paper the word "acoustic" is used. In these two instances it is instead "acoustical". I do not think this is to indicate any semantic difference. Perhaps you might want to keep it consistent. | We have corrected this inconsistency. | The wording has been standardized to use "acoustic" consistently. |

| | | |
|---|---|---|
| 11) 25-27: "during natural disasters" sounds like the kites are in the air while there is a heavy storm. I assume, you mean something like "following I will schedule some time for us to connect. in the aftermath of natural disasters"? | Thank you for the suggestion. | The phrasing was revised to "in the aftermath of natural disasters" for clarity. |
| 12) 27: I suggest "lower mass" instead of "mass savings". Also, it should be stated what AWESs are compared to. | Thank you for the suggestion. | The sentence now uses "lower mass" and specifies the comparison to conventional wind turbines. |
| 13) 33: This is the first time the abbreviation AWES is used. It is introduced a few lines later on l.41. | We appreciate that you noticed the mistake. | The abbreviation AWES is now introduced upon its first mention in the Introduction. |
| 14) 55: I suggest "enhance" instead of "enhances" as it refers to multiple aspects. | Corrected as suggested. | The verb has been revised to "enhance" to match the plurality of aspects discussed. |
| 15) 155-156: I suggest "in the analyses this variable was not considered as a confounding factor". Please clarify that this refers only to experiences with AWE | Indeed, only experience with AWES sounds was not considered as a confounding factor. | The phrasing in Section 2.3 has been revised to clarify that experience with AWES sounds was |

| | | |
|---|---|---|
| sounds as familiarity is considered in later analyses. | | not considered a confounding factor. |
| 16) Figure 3: There is no indication what the color grading represents. | We apologize for the confusion. The original label got covered up during formatting. | The layout of Figure 3 was updated to ensure the color grading label is visible. |
| 17) 230-231 "Examining the spectrograms…": It is suggested that this refers to spectrograms of other recordings than the ones displayed here. If so this should be clarified. | Thank you for the feedback. We didn't mean to imply spectrograms other than the ones displayed and discussed. | The sentence was revised in the Results section to clarify the intent. |
| 18) 251-252: Is the order of SPL below 2 kHz correct? Judging from Figure 4 the levels of C1 look to be higher than those of C2. | We changed 2 kHz to 1.6 kHz to be more precise. Indeed, C1 presents higher levels than C2. The text has been corrected. | The text was updated to reflect the corrected SPL order and specify 1.6 kHz. |
| 19) Description of Figure 6: Although 5th percentile seems to be a standard denotation, next to the explanation of the 25th and 75th percentiles I find it quite irritating as both naming conventions count from different directions. Indicating that the highest | We understand the source of the confusion: The 5th percentile values are the values exceeded 5% of the signal time, as explained in section 2.4.1. | The figure caption for Figure 6 now clarifies the interpretation of the 5th percentile values. |

| | | |
|---|---|---|
| 5% are highlighted might avoid confusion. | | |
| 20) 305: There seems to be a mismatch between text and table regarding the %HA. | Thank you for pointing this out. There were indeed some mistakes. | The text and table values were corrected to ensure consistency in %HA reporting. |
| 21) Figure 7: In some boxes the central horizontal line is hard to see. I suggest the lines and frames of the boxes be black rather than match the color. | A good point, thank you! | The formatting of Figure 7 was updated with black lines for better clarity. |
| 22) 369-370: There is a contradiction to the results chapter. Here it says that the fixed-wing kites show higher loudness, whereas in the results the soft-wing kite had the highest loudness values. | We have corrected the text to reflect that the soft-wing kite had the highest loudness values. | The text was corrected in the Discussion section to match the findings in the Results chapter. |
| **Reviewer 2** | **Response to reviewer 2** | |
| 1) While the literature review in the introduction is short and concise, it would benefit from a greater engagement with more critical research on the assessment of wind turbine | Thank you for your thoughtful comment. In response, we have expanded the Introduction to engage more deeply with research on the social acceptance of wind turbine noise, emphasizing the broader context of noise annoyance and its connection to psychosocial and | The Introduction was expanded to include references to key studies on wind turbine noise and social acceptance. The Discussion section now provides a |

noise embedded in the social acceptance literature: For example Haggett (2012) 'Social experience of noise from wind farms'; Dällenbach & Wüstenhagen (2015) 'How far do noise concerns travel'; Kirkegaard et al. (2025) this journal. The present study seems to divert from the bulk of previous research dealing with noise annoyance and seems to (some extent) speak to this literature not only by focussing on the initial design of an emerging technology, but also by innovatively unpacking the sound quality beyond the loudness of wind turbine sound. It would therefore also be worthwhile to read more about the extent to which the study design, esp. sound quality metrics, have been previously applied in researching noise annoyance from conventional wind turbines and if so, how the findings differ.

individual factors. This expanded discussion highlights the complexity of noise perception and its relevance to the acceptance of wind energy technologies.

We also strengthened the rationale for focusing on sound quality metrics (SQMs) in the context of AWESs by underscoring the innovative contribution of this approach. Unlike conventional studies that primarily emphasize overall loudness, our study seeks to "unpack" the perceptual dimensions of sound, such as sharpness and tonality.

Furthermore, in the revised Discussion section, we provide additional context on how our findings diverge from or align with existing research on wind turbine and drone noise.

comparative analysis of the findings relative to existing wind turbine and drone noise research.

| | | |
|---|---|---|
| 2) Although the description of the methodology provides a lot of detail, it is not entirely clear how sound quality metrics were employed in the study. In the text following line 160, it says that the five metrics were calculated for each sound wave of every recording and it is described how each metric has been calculated. However, it remains unclear how these metrics relate to the participants' perception of the sound emissions, i.e. if or to what extent participants were knowledgeable about these metrics or had the possibility to engage with or refer to such characteristics of sound quality when articulating their annoyance on the verbal scale. Lines 297-299 hint at some interesting terms that describe different degrees of these characteristics, but there is no indication of how these terms have been applied in the participants' assessment and rating of AWES noise. This should be clarified. Similarly, I | We appreciate your comments and the opportunity to clarify the methodology. The study did not assume that participants were knowledgeable about specific SQMs, such as sharpness, loudness, or tonality, nor were they informed of these metrics during the experiment. Instead, these metrics were derived objectively from the acoustic analysis of the recordings. The intention was to identify whether these objectively measured characteristics could predict participants' subjective annoyance ratings (see Section 2.2.3). We have added this explanation at the end of the Introduction, so that readers unfamiliar with SQMs immediately grasp the study rationale.

The descriptive terms mentioned (e.g., "harsh" and "beating") were used to interpret the results of the acoustic analysis and were not terms provided by participants during the experiment. We acknowledge that this distinction could be clearer in the manuscript and have added a sentence at the end of Section 2.4.1, "Acoustic analyses," stating that such terminology arose from the researchers' analysis rather than participant feedback.

Regarding your last point, we want to clarify that participants provided a single annoyance rating for each | Clarifications were added to the Introduction and Section 2.4.1 to explain the relationship between SQMs, participants' subjective ratings, and the study methodology (i.e., use of descriptors). |

| | | |
|---|---|---|
| wonder whether participants' evaluation of individual recordings (line 121) allowed for the possibility of linking their expressions of particular levels of annoyance with certain events in the sound wave. | entire recording, rather than for specific segments within a recording. As a result, our analysis focused on comparing how sound recordings with relatively higher or lower values for certain SQMs (e.g., sharpness, loudness, or tonality) were rated in terms of overall annoyance across recordings. This approach allowed us to investigate whether specific acoustic characteristics of the recordings, as quantified by SQMs, influenced participants' average annoyance ratings. However, because participants evaluated the recordings as a whole, we did not assess their annoyance in relation to specific events or changes within the sound wave of an individual recording. We hope this clarifies the methodology, but we are happy to provide further details if needed. | |
| 3) The conclusion is very short and compact, merely summarising key results and briefly addressing the objective related to the metrics predicting annoyance. A discussion would therefore benefit from a more thorough engagement with the existing literature and much more detail on the significance and wider implications of | Thank you for your valuable feedback. In response, we have added a comprehensive Discussion section to provide a more thorough engagement with the existing literature and to contextualize the study's findings. The new section summarizes the key results and compares them to prior research on wind turbines and drones. We also address the limitations of our findings, particularly regarding their applicability to real-world settings. Furthermore, we outline specific recommendations for | A new Discussion section was added. It summarizes key results, compares findings with prior research on wind turbines and drones, and addresses the study's limitations regarding real-world applicability. The section also includes specific |

| | | |
|---|---|---|
| the findings for the (future) role and utilisation of different airborne wind systems, but also people's acceptance of novel wind energy technologies (or designs) and annoyance of wind turbine sound. In addition, a few more words about the relevance and potential replicability of the lab experiment in a field study would be highly interesting. A greater contextualisation of the study and its findings would be necessary to strengthen its purpose and value. Otherwise, the interesting insights and novel findings would remain rather abstract. | future research and derive actionable insights for industry, emphasizing design modifications and noise mitigation strategies to enhance the social acceptance of AWESs. These additions aim to strengthen the study's purpose, highlight its novel contributions, and demonstrate its broader implications for the development and acceptance of AWE. | recommendations for future research and industry. |
| **FURTHER COMMENTS:**

4) It would be helpful for the reader to better understand and interpret the magnitude of and annoyance of sound emissions from AWES by having a comparison to those of other sources, such as conventional wind turbines or drones. How comparable are the | Thank you for this suggestion. While direct comparisons to other noise sources were beyond the scope of this study, we agree that providing context for the observed annoyance levels is helpful. We have added a comparison with two studies that assessed annoyance for wind turbine sound in laboratory settings under similar conditions. Specifically, we note: "The mean annoyance ratings for the different AWES types ranged | Comparisons with existing research on wind turbine noise annoyance were added to the Results section to help contextualize the annoyance ratings. |

| insights gathered by drawing on sound quality metrics of AWES? | from approximately 34 for AWES A to 54 for AWES C (Fig. 7). In comparison, Merino-Martínez et al. (2021) reported average annoyance ratings of about 61-72 (converted from the ICBEN 11-point scale to a 0-100 scale) for wind turbine sound in a laboratory experiment, although the LAeq values in their study were lower, at around 38 dBA, than those used here. The percentage of highly annoyed participants (%HA) per AWES type varied between approximately 1% and 19% (Table 4), with AWES C showing the highest %HA, followed by AWES B and then A. This trend aligns with the previously reported higher tonality and sharpness values for AWES C compared to B and A. The observed %HA range  is slightly narrower than the 2 to 34% of HA predicted by Schäffer et al (2016) for wind turbine sound exposure in laboratory settings with an LAeq range of 35 to 45 dBA."

Unfortunately, we couldn't find more relevant studies because existing research on wind turbine noise annoyance varies significantly in methodology, including differences in annoyance scales (e.g., 5-point vs. 11-point), thresholds for defining "highly annoyed," sound pressure levels, and study settings (laboratory vs. field). Regarding comparisons with other studies on | |

| | SQMs, we address this point in response to your first comment. | |
|---|---|---|
| 5) Line 126: The definition of annoyance is not entirely clear or at least a bit infelicitously formulated. First, it is tautological, as annoyance is defined by referring to the term annoyance in its explanation (line 127). Second, it seems as if the defined term should be 'noise annoyance' rather than annoyance. Thus, I recommend to rephrase the definition. | We appreciate your observation and agree that the definition can be clarified to avoid tautology and better align with the term "noise annoyance." | The definition of "noise annoyance" was revised to align with the ISO 15666 standard, providing a clearer and more precise explanation. |
| 6) Line 142: it would be useful for the reader to see a list of 'items covering noise sensitivity in different situations' | We agree that including a few sample items from the NoiSeQ scale would improve the clarity and comprehensiveness of the manuscript. However, we have opted not to list all the items to avoid unnecessarily lengthening the text. | Sample items from the NoiSeQ scale were added to illustrate how noise sensitivity was assessed. |
| 7) Lines 153-155: Having a sample where roughly 50% are familiar of airborne wind seems unusually high. In turn, the fact that only few participants have heard the sound of an AWES | We value your feedback and agree that participant familiarity with AWES warrants further elaboration. The higher level of familiarity with AWES in our sample can be attributed to the presence of a highly active and renowned research group on airborne wind energy at | Text was added to the Results section discussing potential influences of participant familiarity and attitudes and it was clarified in the Method why half the sample |

before would presumably be even more pronounced in a field experiment, given the novelty of the technology and the fact that only very few people have seen an AWES and have been consciously exposed to its sound. This raises questions about the familiarity of participants with AWES. How does the recruitment of younger participants from a technical university affect the familiarity with AWES and attitude to AWES? Would the attitude be different in a random sample that includes less familiarity? It would be useful to critically consider these issues here or in the discussion/conclusion.

our faculty. Many participants are students or employees and may have been indirectly exposed to the concept of AWES through institutional activities or research dissemination. However, it is important to note that this familiarity was primarily theoretical. Given the limited number of operational test sites worldwide, most participants had never encountered an AWES in real life, seen detailed footage, or heard the sound emissions prior to the experiment. This explains why only a small number of participants had previously heard the sound of an AWES. We added this explanation in Section 2.3 "Participant characteristics".

We acknowledge that participant familiarity may differ in a random sample. However, although the proportion of people familiar with AWE is likely higher in our sample than in the general population, there was still an even split between those familiar and unfamiliar in the sample. This balance allowed us to compare the effect of familiarity on annoyance ratings within both the familiar and unfamiliar subgroups within a population that may generally be more exposed to such technology. Moreover, the main effect of familiarity on annoyance was statistically insignificant, suggesting that familiarity alone did not significantly influence annoyance ratings. Regarding field studies, they

was familiar with AWE. The Discussion section now includes notes on sample representativeness, particularly age-related effects, and suggestions for future research.

| | typically focus on individuals who have been exposed to the sound emissions of the technology in question. Familiarity with AWES, in this context, would likely be a prerequisite for obtaining meaningful results regarding real-world noise annoyance. | |
| :-- | :-- | :-- |
| | The first reviewer also noted a potential entanglement of familiarity with attitudes in that participants' familiarity with AWESs may reflect a more positive attitude, leading to lower annoyance ratings. To address this point, we have added the following explanation in the Results: "This familiarity may be intertwined with more positive attitudes toward AWE, potentially explaining the lower levels of noise annoyance observed—a pattern consistently reported in studies on wind turbines." | |
| 8) Lines 283-285. This sentence requires more explanation. It is not clear why C3 is obviously different from C1 and C2 (Fig4). How is the higher loudness of C3 related to the 'frequency-dependent human ear sensitivity'? Does this mean that the | In Figure 4c, the spectra of C1, C2, and C3 are plotted. A noticeable difference in sound pressure levels (Lp) is observed for frequencies below 1.6 kHz, particularly for C2, which exhibits lower Lp compared to C1 and C3. Additionally, within this frequency range, C3 elucidates peaks that suggest a tonal behavior. Human hearing is most sensitive to frequencies between 2 kHz and 5 kHz, as illustrated by the equal-loudness contours (ISO 226). | Text was added to the Results explaining the higher loudness values of recording C3. |

| | | |
|---|---|---|
| recording of C3 recorded a louder sound of the reel-out phase? | Observing Figure 4c, it can be noted that C3 displays higher Lp values in the frequency range of 3–5 kHz. | |
| 9) Lines 297-299. With reference to my comment above, the terminology highlighted in these two sentences seems to be crucial for qualifying the different perception-based sound metrics (e.g. harshest, strongest beating ...), but it is not obvious where these terms are coming from and how they have been applied in the study. Were these terms included in the verbal scale, were they articulated or rated by the participants, or were they merely used by the researchers to illustrate the results? | Thank you for raising this point. As explained in response to your earlier comment, these descriptive terms were used to interpret the results of the acoustic analysis and were not terms provided by participants during the experiment. We acknowledge that this distinction could be clearer in the manuscript. | A sentence was added at the end of "Section 2.4.1 Acoustic analyses," stating that such terminology arose from the researchers' analysis rather than participant feedback. |
| 10) Are there any further limitations in the study that should be mentioned? Are there any challenges concerning the recordings, considering the distance to the microphone and moving kites (also compared to the recording of the sound from conventional turbines)? | Thank you for your suggestion to add a limitation section. We have done so. The following part pertains to the aspects you mentioned: "[…] the study's methodology faced several challenges related to sound recordings, particularly concerning the varying distances to the microphone (100–700 m) and the moving nature of the kites. While normalization to 45 | A limitation section was added to discuss challenges related to sound recordings and other factors, including distance to microphones, moving kites, and environmental factors. The restricted applicability of |

| | | |
|---|---|---|
| | dBA mitigated some inconsistencies, the dynamic sound signatures created by the kites' flight patterns introduced additional variability compared to the noise emissions of stationary wind turbines. Additionally, the location of the observer or microphone significantly influences noise perception because the acoustic prominence of different system components varies depending on the vantage point. For example, certain components, such as the kite, may dominate acoustically when the observer is positioned directly below or in line with the kite's trajectory. In contrast, noise from the generator or tether vibrations may become more prominent at close distances to the ground station. Furthermore, environmental factors such as wind noise and ground reflections may have influenced the recordings despite mitigation efforts using windscreens. These limitations underscore that the results have only restricted applicability to the field. Schäffer et al. (2016) highlighted these challenges, emphasizing that laboratory and field studies should be viewed as complementary rather than directly comparable." | laboratory findings to field settings was emphasized. |
| **MINOR COMMENTS:** | | |

| 11) Line 27: If mentioned, it is perhaps useful to elaborate a bit more on mass savings and how they relate to capacities and energy production of AWE in comparison to conventional wind energy? | The first reviewer also wondered what AWESs are compared to here, so we revised the sentence: "With its substantially lower mass compared to conventional wind turbines, AWE also has a smaller environmental footprint." For the sake of conciseness, we refrain from giving more details in the Introduction but refer you and other interested readers to the following two sources for more information: Hagen et al. (2023) found that future large-scale hard-wing AWE systems deployed in onshore wind parks could save 70% of the mass of a comparable HAWT park. The global warming potential (GWP) and cumulative energy demand (CED) were 60% and 65% of the respective impacts of the HAWT. Coutinho (2024) investigated mobile soft-wing AWE systems and found that the mass savings were also significant but less pronounced. Similarly, the GWP and CED were reduced less than the HAWT installations. | Phrasing in the Introduction was revised to clarify the lower mass and environmental benefits of AWESs compared to conventional wind turbines. |
| --- | --- | --- |
| 12) Line 28: Explain how and why AWE is seen as a potential game-changer in the energy transition. | We appreciate your suggestion and have expanded on this point to clarify why AWE is considered a game-changer. | The Introduction was expanded to explain why AWE is seen as a potential game-changer, with points on harnessing high-altitude |

| | | winds, material efficiency, and deployment in remote or offshore locations. |
|---|---|---|
| 13) Line 61-62: It would be interesting to know what AWES was used at this test site, fixed-wing or soft-wing. | Thank you for pointing this out. | Specified in the text that a soft-wing AWES was used at the test site. |
| 14) Line 65: It may be better to write 'sound emissions of different AWES' instead of 'AWES sound emissions' in order to hint at the scope of your study | The sentence in question refers to the field study by Schmidt et al. The study only investigated one type of AWES. Therefore, we will retain the phrasing. | No changes were necessary in response to this comment. |
| 15) Line 69: What is meant by global in global Psychoacoustic Annoyance (PA)? The term global is not used again in the remainder of the manuscript. | We appreciate your attention to detail. The term 'global' in Global Psychoacoustic Annoyance (PA) refers to a composite metric that integrates multiple sound quality metrics to estimate annoyance, as explained. | Removed the redundant use of the term "global" in Global Psychoacoustic Annoyance (PA) for consistency and clarity. |
| 16) Line 89: Normalisation and A-weighting should be explained for the unfamiliar reader. | We agree. | Added brief explanations of normalization and A-weighting in Section 2.4 to aid readers unfamiliar with these terms. |

| | | |
|---|---|---|
| 17) Line 109: the caption of the figure requires a source, even if it is an own photograph | Thank you for the suggestion. | A source was added to the figure caption, specifying that it was an original photograph. |
| 18) Line 138: 'reasonably valid' sounds more like guesswork rather than scientific severity | Thank you for pointing this out. | Revised the phrasing to replace "reasonably valid" with a statement emphasizing scientific rigor regarding self-evaluations of hearing ability. |
| 19) Line 152: the information about the duration of each experiment could be moved forward to section 2.2.2, as I was wondering about that while reading this section | We agree. | Relocated the duration of experiments to Section 2.2.2 for better logical flow. |
| 20) Line 160: the verb 'describe' annoyance is used here, whereas the term 'predict' annoyance is predominantly used later on (e.g. lines 355-362). Is there a difference in the meaning? If not, it might be preferable to use one verb consistently in order to avoid confusion for the unfamiliar reader | Since "predict" is statistically more precise, we will use it consistently. | Standardized the use of "predict" instead of "describe" when referring to the statistical role of SQMs in annoyance ratings. |

| | | |
|---|---|---|
| 21) Line 175: Di et al. (2016) does not seem to appear in the bibliography | Thank you for noticing. | The missing reference to Di et al. (2016) was added to the bibliography. |
| 22) Line 335-336: Elaborate a bit on this sentence. What does 'varying effects on annoyance rating among individuals' mean? | We have clarified this point about how individual differences (e.g., noise sensitivity, age, and familiarity with AWES) could influence the perception of specific sound qualities. | The text was revised to clarify what is meant by "varying effects on annoyance rating among individuals." |
| 23) Lines 355 and 359: add years to references and 'et al.' to Di | We appreciate the observation. | These citations were corrected to include the publication years and the proper "et al." formatting. |